# Bibliometric Analysis on the Implementation of Evidence-Based Practices through Building Effective Systems

**DOI:** 10.3390/children10050813

**Published:** 2023-04-29

**Authors:** Julia Argente, Gabriel Martínez-Rico, Rómulo J. González-García, Margarita Cañadas

**Affiliations:** 1Doctoral School, Catholic University of Valencia San Vicente Mártir, 46001 Valencia, Spain; 2Campus Capacitas, Catholic University of Valencia San Vicente Mártir, 46001 Valencia, Spain

**Keywords:** early intervention, implementation science, system framework, evidence-based practices, recommended practices

## Abstract

Implementing evidence-based practices in early intervention has generated new research interest in the need for effective early childhood systems. This study used a bibliometric analysis to discern the importance and relevance of the field. The analysis highlights that the main goals of the future direction of the research field need to be better defined. First, articles published in the Web of Science database between 2012 and 2022 were reviewed, and then word and author combinations were analyzed. Finally, articles were collected in different groups for bibliographic linking. Five key points were identified as the most important practices recommended by the Department of Early Childhood or Professional Development. It should be noted that the main difficulty encountered in this field arises from the novelty of our research topic, i.e., there is no research on constructing systems for early intervention. However, it is worth highlighting the articles that are relevant to the field of study and our success with integrating them in order to demonstrate the importance of serving children with disabilities and their families. In conclusion, the establishment of a system built on evidence-based practices is underdeveloped but shows promise for the future of early intervention systems.

## 1. Introduction

Over the last decade, research has been conducted worldwide on the importance and significance of implementing evidence-based practices in programs for children and their families [1]. Researchers refer, on the one hand, to emerging theories in child development and, on the other hand, the science of implementation.

The ecological model is one emerging standard from developmental science. The ecological model insists that there are three layers in the immediate environment (family, school, and social field), where interconnections are established between the different elements of the immediate environment and the last layer, which involves the value system of one’s society (traditions, policy, and religion). Therefore, we must consider all layers when providing services for children with disabilities and their families, from the immediate environment and the interactions that are created in it to the policies and infrastructure that will impact the quality of the services they receive [2]. Along the same lines, the evidence stresses the importance of the system and the social units that have a direct or indirect impact on providing early intervention services [3]. The theory of system implementation in early childhood care integrates the perspective of the theoretical model (family, school, or social field) in its planning [1,4,5].

Moreover, implementation science in early intervention consists of establishing a plan of action that involves scheduling activities for all phases of the process. Specifically, having administrative, professional, and family structures to convert the evidence and the policy ideas into practice is essential to achieving the best results for children with disabilities and their families [4]. The implementation of evidence-based practices in early intervention entails a broad process that requires the active participation of the agents involved: social, educational, health, family, and state services. Moreover, it requires a systematic organization to guide the development of the process, monitor its phases, and apply different quality and evaluation indicators rigorously [5].

Authors such as those of [6] tend to believe that both theories stress the importance of policy and finance, among other important elements, to deliver services based on scientific evidence. Although the researchers do not say so directly, we can highlight the importance of building efficacious systems to obtain the best results for children and their families as supported by scientific evidence.

That is why the Early Childhood Technical Assistance Center created a systems framework in 2015 after undergoing a rigorous two-year development process underpinned by these two theories. This methodology states that an effective system is made up of a set of interdependent elements [7,8] that ensure governmental, financial, and professional development support for evidence-based practices [6]. The system, in addition, integrates continuous evaluation, the development of an improvement plan, and the stability of services [9] to ensure the achievement of the expected results in each of the phases of implementation. In this sense, we can affirm that the construction of systems makes possible the methodical incorporation of scientific evidence into organizational practices and early childhood care networks [10] from the rationale of the two theories developed above.

Researchers argue for the need to comprehensively analyze the implementation and system construction processes in the international context. Accordingly, the present study performs a bibliometric analysis of the implementation of evidence-based practices through the construction of effective systems [11]. As a result, our study aims to provide scientific evidence on this important issue by identifying the most relevant characteristics and aspects related to the construction of systems (topics treated, research groups, evolution, relevance of the subject matter, and main limitations).

Several recent bibliometric analyses on the implementation of evidence-based practices show that it is a theme that has evolved over the last decade [12,13,14,15]. In addition, this study aims to examine articles published on Web of Science related to the construction of this first intervention system. In addition, this requires the use of an evidence base to achieve the best outcomes for children with disabilities and their families. The extension of this type of analysis is important because the field of early intervention and early childhood system building is expanding, which coincides with the study of the implementation of evidence-based practices. This has stimulated research on strategic systems that are now of great importance for the management interventions of government organizations [5,16]. The following is one of our conclusions: this type of analysis is interesting as it builds a bridge between theory and practice in terms of the implementation of evidence-based practices. This will help government organizations build high-value systems to provide the best outcomes for children and care for their families [5,17].

Through a bibliometric analysis, this study aims to explore the current state of research on system building by applying an evidence-based approach to early intervention. To achieve this goal, this study was guided by the following research questions based on implementation science:RQ1: How has the publication of articles related to system building from the implementation of evidence-based practices in early intervention evolved?RQ2: Which authors have produced the most publications related to system building since the implementation of evidence-based practices in early intervention, and which of these authors’ work is most mentioned?RQ3: Which journals have produced the most research on the topic of building systems through the implementation of evidence-based practices in early intervention, and what are their Impact Factors?RQ4: What co-authorship networks, co-citations, and co-words are associated with studies detailing system building from the implementation of evidence-based practices in early intervention?RQ5: What are the main topics to be explored in this research?

## 2. Materials and Methods

### 2.1. Data Collection

Bibliometrics uses bibliographic data to analyze the most influential articles in a particular field of study. Specifically, we looked at all published articles indexed in the Web of Science Core Collection™ (SSCI, SCI-Expanded) that are concerned with building evidence-based systems based on practices in early intervention. Only Web of Science (WoS) publications were considered. This is because it is the most authoritative database for collecting and analyzing scientific papers [18,19,20]. In addition, a great amount of research has been carried out in specific studies using three indicators: abstract, title, and main chapter. Therefore, the method described in [21,22,23] was adopted. The most important words in the content of an article were placed in different sections. Therefore, the examination line used in the topic field was ((early + intervention * OR childhood) AND (implementation OR implementation OR implementation + science OR implementation + practise) OR (system + improvement) OR (recommend + practise)). These terms were chosen to keep in mind our research objective of addressing research gaps in system building through the application of evidence-based practices. The results of this bibliographic analysis suggest that articles published in influential journals have found a common framework for building systems that work in early intervention by implementing evidence-based practices. All these terms are associated with positive outcomes for children with disabilities and their families. Therefore, language modes were not included in the bibliographic analysis of the texts. However, English was the main language in our research.

Additionally, since databases are constantly changing and improving, it is important to demonstrate the longevity of our article collection [24]. The search was limited to review publications and articles and excluded results such as book chapters, book editorials, and anthologies. The first step involved finding documents from 1823 to 2022. We followed the procedure for removing articles that are not related to the topic under study (eligibility). The authors adopted the PRISMA (Preferred Reporting Items for Systematic Reviews and Meta-Analyses) framework to ensure transparency and rigor during the review process [11,22]. During the screening process, inappropriate papers were excluded (*n* = 1337). In the third step, we checked the relevance of the remaining papers by evaluating their titles, keywords, and abstracts. (1) Papers were excluded if they discussed early intervention but did not report the implementation of best practice system building or did not present a specific intervention model. After this process, 128 articles were included in the final database analysis. These data are presented in Figure 1.

### 2.2. Bibliometric Analysis

After obtaining the plain text data, duplicate and unknown data were identified and homogenized. One of the main challenges we faced was duplicating authors with different name variants. To address this, we reviewed the full article to avoid duplication and errors and to identify some significant distortions of information, such as the address of the institution and year of publication. Therefore, the analysis was conducted in two ways. HistCite statistical software (version 2010.12.6; HistCite Software LLC., New York, NY, USA) was used first. Second, we used techniques such as bibliographic relevance discourse analysis and historical analysis to identify related themes and outline emerging areas of research. We aimed to provide a detailed analysis of the research construction system in the implementation of the evidence-based use of early intervention to analyze the themes present in this field of research [11,22,25]. Common keywords were used to explore common topics that should be developed in the field of intelligence research [26]. We also conducted a bibliographic analysis to identify specific lineages in the field of building systems research from the implementation of evidence-based practices to early intervention. This method measures the similarity between two articles by distinguishing the number of identical references. The advantage of bibliographic union analysis is that the number of ideas cited in a study does not change over time, unlike current analysis, which can be affected by the time of implementation. Thus, bibliographic link analysis provides a more powerful and reliable method for identifying relevant links to relevant articles in a research field [10]. For this reason, the best solution is to complete a systematic literature review, and this approach has been used in previous studies [10,11,22] to correctly interpret the tables produced. Finally, the data obtained were also analyzed using R Studio v.3.4.1 software and R bibliometric package (http://www.bibliometrix.org (accessed on 11 February 2023)) [27]. The data were imported into R Studio and organized into bibliographic tables. This software was used to analyze the basic search string information and to implement analysis tools. The advantage of this is that a table of common tools based on word analysis allows for the identification of research topics and the main research focus of first interventions with construction systems with the implementation of exercises based on topics.

## 3. Results

This section is organized into subsections to provide a clear and concise presentation of the experimental results, their interpretation, and the conclusions drawn from the study. The study’s search in the WoS database resulted in 128 articles published in 49 journals authored by 428 researchers from 17 countries. The mean number of citations per document was found to be 13.11. A total of 318 keywords and 379 author keywords were identified. The average number of authors per paper was 3.34, and the average collaboration index was 3.54. Further details regarding the results of the study are presented in Table 1.

### 3.1. Basic Indicators

This section provides an overview of the basic article and citation metrics and the evolution of the number of articles and citations from the authors’ institutions and national journals. In addition, various indicators including quantity and quality are taken into account. The Global Citation Score (GCS) reflects the total number of citations of analyzed articles in the WoS database. The local citation score (LCS) refers to the number of citations obtained from the WoS database only for the papers selected in this study. Finally, we present the evolution of the authors’ keywords over the years.

#### 3.1.1. Years

The articles produced on the implementation of comprehensive systems in early intervention services have increased over the years. In the last 10 years, there has been an increase in the number of publications on this topic. A turning point can be observed in 2015 (14), and from 2019 there was an increase in the number of papers, (2019 (13), 2020 (17), 2021 (21), and 2022 (24)). According to the number of citations, articles written in 2015 had the highest number of citations (537), followed by 2012 and 2013, respectively. Figure 2 shows the evolution of the number of articles and citations.

#### 3.1.2. Authors

A total of 428 researchers published at least one article on this topic. However, the researchers with an elevated quantity of publications on building systems in early intervention services from the implementation of evidence-based practices or best practices have been McWilliam with seven papers, followed by Stahmer with six papers, Barton with five, and Dunst and Garcia-Grau with four papers each. We can observe that Snyder and Stahmer are the authors with the most global citations, with 226 and 184, respectively. The results are shown in Table 2.

Further, the authors who have received the highest number of citations of their research are Snyder, PA (Nb = 4; GCS = 226), Stahmer, AC (Nb = 7; GCS = 184), Mandell, DS (Nb = 4; GCS = 150), and Dunst (Nb = 4; GCS = 102), who have more than 100 global citations each.

#### 3.1.3. Countries

Concerning scientific publications by country, the USA has a high number of publications (200). Countries such as Australia, Canada, China, and Spain have also published on the subject, but to a lesser extent. This information is shown in Figure 3.

#### 3.1.4. Journals

A total of 49 journals have published at least one article on system building in early intervention, as is shown in Table 3. Among these journals, 17 have published multiple articles on this topic, while the remaining 32 journals have only published a single article and are not included in the table. The journal with the highest global citation score per number of articles (GCS/NB) is “Topics in Early Childhood Special Education” with a score of 19.87, followed by “Infants & Young Children” with a score of 17.5. “Topics in Early Childhood Special Education” has also published the most articles, with a total of 18, while “Journal of Early Intervention” has published 13 articles with a lower GCS/NB of 14.54. The other journals published between eight and two articles on this topic.

In terms of Impact Factors, Autism has the highest Impact Factor (JIF = 6.684; Q2), followed by Early Childhood Special Education Topics (JIF = 2.313; Q2) and Journal of Early Intervention (JIF = 1.925; Q2), respectively. The results are presented in Table 3.

#### 3.1.5. Most Common Keywords

The most conventional and relevant keywords the authors used at any time are presented in Figure 4. The term “early intervention” has been the most searched keyword in terms of the subject matter of this research, with a clear upward trend since 2009. It also highlights the evolution of the term “professional development” in the last decade, with a substantial increase in searches since 2016. It is also worth commenting on the decrease in mentions that has been observed in recent years of terms such as “early childhood education”, “early childhood”, “challenge behavior”, “implementation”, and “governance”. As we can see, the word “systems framework” does not appear, but there are words that are part of this construction of systems.

### 3.2. Co-Citation Analysis

In this second section, we propose an analysis of co-citations. First co-author networks are described. Then, international collaboration networks are described, and finally, keyword networks are defined.

#### 3.2.1. Co-Authorship

A total of 10 networks of co-authorship were identified among 32 prominent early childhood researchers. However, only the collaborations of co-authors of one or more articles out of 428 authors are presented. A group is proposed for each network using the Louvain algorithm. Specifically, there are seventeen researcher networks; three networks with four researchers, one network with three researchers, and five networks with two researchers. We can see that Stahmer and McWilliam, the highly published writers mentioned above, have many collaborations with different editors. In contrast, Dunst, who is one of the most published authors, collaborates less with different authors. Figure 5 shows various collaborative networks.

#### 3.2.2. Co-Word Analysis

There are five main keyword groups. Breakpoints are generated when these keywords occur five or more times. The first consists of 15 words (purple groups) representing the peak and consequent words. The second group includes 12 words related to children (red cluster), and the most important word in this block is “early intervention”.

The third grid consists of nine terms (green clusters) and has professional development as its central term. This demonstrates the effectiveness of the model in early intervention programs. The fourth network consists of seven words (blue clusters) and exercises on children with disabilities. Finally, the fifth group consists of five words (orange blocks) representing young children or behavioral intentions. Figure 6 shows this set.

### 3.3. Thematic Analysis

Finally, this third section presents the thematic analysis. On the one hand, a bibliographic analysis of the combination is provided, and, on the other hand, strategy maps on various topics are provided. Ultimately, all these results are displayed as a map.

#### 3.3.1. Bibliographic Coupling

A bibliographic coupling analysis was conducted. A point interval of at least two citations is attached. After this, only documents that met this criterion were selected, leaving the final analysis with 50 documents in four distinct clusters (one color per cluster). After that, to perform the content analysis, only the most relevant articles from each cluster were selected based on the higher number of citations. Figure 7 shows this set.

##### Blue Cluster (42 Citations, Six Articles): Family Coaching and Capacity Building

This group contains six articles with a total of 42 citations. A highly cited article reviews the scientific evidence and highlights the gap between early intervention evidence and practice. It also highlights the benefits of family learning and capacity building in line with the Department of Early Childhood (DEC) recommended practices that demonstrate the importance of supporting caregiver–child interactions in everyday life. The second-most-cited article explores the benefits and strategies of training caregivers of children with disabilities in early intervention programs, including reflection and problem-solving strategies. Finally, the third article describes family-centered early intervention methods based on the seven thematic areas of the DEC recommended practices published in 2014. It aims to encourage caregiver reflection and guidance in these types of practices. In summary, the three articles in this group work towards better outcomes for children with disabilities and their families, which reflects the importance of capacity building and training for carers. These articles, therefore, suggest that early intervention should follow the DEC’s recommended practice evidence by creating a system through training and home visits by caregivers.

##### Purple Cluster (284 Citations, 12 Papers): Professional Development

This cluster is composed of 12 papers with a total of 284 citations. The topic of this cluster is related to the importance of professional development in early intervention services. The article with the most citations analyzes coaching theories and strategies to support adult learning. Additionally, it indicates implementation differences and what should be involved in professional development in these practices. This is followed by another article that argues for the implementation of science in early intervention to promote intervention practices based on data from a study on adult learning. The third article insists on a professional development design during the promotion of coaching caregivers and adult learning principles. The last article demonstrates the efficacy of recommended practices with a multiple-baseline design, emphasizing the use of caregiver coaching strategies via performance feedback by email.

##### Red Cluster (62 Citations, 14 Papers): Family-Centered Practices

This cluster is one of the largest and consists of 14 papers. There are 62 citations in total. The contents of these documents describe how to implement a family-centered approach and the benefits and advantages of such an approach. The most-cited article in this cluster focuses on how to implement a family-centered model in early intervention [28] and has been cited 21 times. Its authors support the approach of implementing a family-centered model in early intervention through quantitative and qualitative analysis. Therefore, this article describes a program that uses professional practice to enhance the developmental capabilities of children in families. It describes a 10-country expert early intervention model with a family-focused but unique approach: routine interview-based and support-based visits with a focus on family and child involvement. In addition, Part C of the article, Parents Experiences in Rural Areas: Alignment with Recommended Practices [29], analyzed recommended practices in rural areas by interviewing parents to determine whether the practices used were consistent with recommended practices. Finally, Early Intervention Practice in Southeastern Spain: Professional and Family Perspectives and Evaluation of Early Interventors examines practices and beliefs and makes suggestions for improvement.

##### Green Cluster (275 Citations, 17 Papers): Early Childhood Special Education

At the fifth level, the green cluster consists of 17 articles with a total of 257 citations. This group or series of articles explores research on early childhood practice and leadership in the delivery of services to children and their families. The most-cited article, which supports the implementation of evidence-based practices in practice-based training [30], has 140 citations. This article describes a training framework designed to help early childhood education professionals improve teaching practices and adhere to evidence-based practices. This cluster also included Leadership in Implementing Quality Family-Centered Services in Early Childhood: Exploring Administrators Understanding of Needs and Realities [31] and Implementation of Mixed Approach Early Intervention in Pennsylvania: Supportive Services and Policies for the Growth and Development of Children with Disabilities [32]. The first of these analyzes the impact of using recommended practices in early childhood classrooms. The second concerns the use of perspectives and recommendations in a qualitative study of the problems faced by family-centered implementation. The third section focuses on high-quality early intervention programs and analyzes interventions for children and their families that support public policies that aim to foster professional development and family engagement.

#### 3.3.2. Strategic Thematic Analysis

Below is a diagram of the strategies used in the field of building systems in early intervention services by implementing evidence-based or best practices with positive outcomes for children with disabilities and their families. The dot size corresponds to the number of occurrences of these keywords. The upper left quadrant contains more specialized/specific topics such as affective learning and collaborative therapy recommendations related to this field of study. In contrast, the upper right quadrant contains sensitive themes, including well-developed internal relationships, but few important external relationships, and thus has more important perspectives for the child and interventions in the child’s behavior. However, due to its centrality and scale, the child’s perspective is likely to emerge as a driving theme in the coming years. Themes in the lower left quadrant represent emerging or disappearing themes that are undeveloped and relational. Quality seems to be lost in the map area, for example. Finally, the topics in the lower right are fundamental to this field of study but are still evolving. This quadrant includes basic interdisciplinary and general topics such as early intervention models and their implementation in young children. Figure 8 shows this.

## 4. Discussion

In recent years, implementation science has allowed us to improve early intervention services, providing evidence about the steps required to implement evidence-based practices to improve outcomes in services for young children and their families [5]. Due to the scientific evidence provided by implementation science, there is increased interest in generating effective systems composed of a set of elements, all of which are necessary to complete the expected outcomes for children and their families [33]. It is important to emphasize that each element has a specific role and activities that influence and impact the other components, but all of them must work together effectively to provide a high-quality system [34].

However, despite the efforts of professionals, managers, boards of directors, and families of the entities, among others, the data make it clear that attaining the best outcomes for children and families requires an implementation plan starting with building statewide systems. This system must ensure governmental, financial, and professional development support for evidence-based practices, as well as ongoing evaluation, the development of an improvement plan, and the stability of services [7,35].

The field of study of systems building through the implementation of evidence-based practices is becoming an area of study within the field of early intervention and service transformation to decrease the gap between research and practice [5,36].

We obtained clarifying data from our thematic analysis and the different clusters. There is a research focus on “early intervention”, “professional development”, “family-centered” and “care collaboration consulting”, “early childhood”, and “implementation”, which are essential topics in the construction of systems through the implementation of evidence-based practices, but this shows us that they are still to be developed [13].

Based on our analysis, no research develops the construction of systems through the implementation of evidence-based practices in their entirety, but we do find all the necessary steps in the publications as a whole [9]. In addition, it is worth mentioning that although the legislative framework is driving the construction of systems in the early care system, and recognized entities [8] are putting into place the means to implement such systems, this study shows that there is a lack of system building related to evidence-based practices [37].

This bibliometric article certainly allows us to see the importance of publications that bring together all the elements that a system needs to be able to carry out the adequate implementation of evidence-based practices to produce the best results for children and their families [38].

RQ1: How has the publication of articles related to system building from the implementation of evidence-based practices in early intervention evolved?

The small number of articles on this topic is consistent with the relatively small number of articles (128) that explore system building rather than implementing evidence-based practices. This could be because, despite being a topic that is reviewed a lot, researchers do not gather the six elements (Governance, Finance, Personnel, Data systems, Accountability, and Quality Improvement and Quality Standards) of the system together. Along the same lines, the elements are analyzed in an isolated way, making synergy between them difficult, meaning that the expected results in early intervention services are not obtained.

RQ2: Which authors have produced the most publications related to system building since the implementation of evidence-based practices in early intervention, and which of these authors’ work is most mentioned?

Although there are many authors with publications, most of them have published fewer than four articles related to the topic at hand. There are five authors who have published more papers. McWilliam has published seven articles, one more than Stahmer, but Stahmer has more citations (GCS = 184) compared with McWilliam (GCS = 41). Dunst (GCS = 102) has published four articles, as has García-Grau (GCS = 27), but he has more citations than McWilliam or Barton (GCS = 71), who has five publications but fewer citations. Dunst does not feature among the authors with the uppermost number of publications from 2013 to 2019, but he is an important author in terms of the subject that concerns us [39,40].

RQ3: Which journals have produced the most research on the topic of building systems through the implementation of evidence-based practices in early intervention, and what are their Impact Factors?

Thirty journals have published more than one article on system building from the implementation of evidence-based practices, and twelve journals have published more than two articles. Among these journals, “Infant & Young Children” is the one with the highest number of published articles (18) and the highest global citation score (315), but it does not have the highest Impact Factor. It is followed by “Topics in Early Childhood Special Education”, which has a total of 15 published articles and 298 global citations. The journal with the highest Impact Factor is Autism (6.684).

RQ4: What co-authorship networks, co-citations, and co-words are associated with studies detailing system building from the implementation of evidence-based practices in early intervention?

We found 10 co-authorship networks with a low level of collaboration among them. Of the total of 428 authors, we only found collaborations between authors who have written one or more articles together. We should highlight Stahmer and McWilliam, who have engaged in collaboration with many different authors, in contrast to Dunst, who is the author with the fourth-highest number of publications but very seldom collaborates with different authors.

Four main groups were identified according to the groups of keywords. One of them is compared with first responders and portrays different models and their effectiveness. The second network focuses on interventions that encourage the development of professionals to provide better services to children and families. In the third, the effect of network science is utilized for experience-based practices and interventions. Finally, the fourth model involves domestic network installation.

The number of publications in books, journals, and documents (and even unpublished works) worldwide, although significant, is low when considering the population that could potentially benefit from early intervention; the variety, quality, availability, and organization of the services; as well as the training of human resources required to make it possible.

RQ5: What are the main topics to be explored in this research?

Four main clusters have been found in building systems through the implementation of evidence-based practices. Of these, the DEC recommended practices cluster stands out as being the most commonly cited, which analyzes the scientific evidence of the effectiveness of evidence-based practices, reviews it quantitatively and qualitatively, and suggests steps to be taken for future research.

The second-most-cited cluster is the one that refers to the implementation of practices in families referring to what type of coaching strategies can be given to families, as well as the importance of the professional development involved in the intervention of children with disabilities and their families. This cluster reflects the importance of the construction of systems that involve all the people responsible for the care of children with disabilities and their families working in a coordinated manner.

The third cluster, which has many citations, is related to intervention and practices carried out in early childhood, leadership in the implementation of these practices, and the gap between the research on recommended practices and current interventions. Additionally, it analyzes early childhood services by analyzing the different perspectives of the varying intervention models and the need for the implementation of science and practices to obtain the best outcomes for children and families.

The fourth cluster refers to family-centered practices and examines the implementation and benefits of this type of practice. The different articles analyze both the perception of professionals and families, as well as the theories and policies for carrying out this type of practice, which are elements in the construction of early intervention systems. This cluster, with only 42 citations, indicates the importance of evidence in early intervention practices, insisting on the recommended practices and strategies of family coaching.

## 5. Theoretical and Practical Implications of this Study

The field of the implementation of evidence-based practices through building effective systems is becoming an area of study in early intervention, reducing the gap between research and practice [5,36,41].

Improving early intervention services requires an implementation plan that is based on scientific evidence, as the practices are to be transformed. Thus, implementation science establishes phases and requirements for implementing evidence-based practices, along with ecological theory, which defines the existence of a functioning early intervention system (exosystem) as a key component that is aimed at obtaining the best outcomes for children with disabilities and their families. Even though implementation is not an isolated process, common criteria are needed to ensure the implementation of evidence-based practices [42]. It is worth highlighting the lack of research on the influence of systems and the use of frameworks to support the improvement of early intervention services [6].

In brief, we certainly need more research on this topic to know all the components that are needed to conduct an adequate implementation of evidence-based practices in depth. Therefore, we consider that this article, unlike others, analyzes the research and justifies the importance of continuing to advance this field of study, which will undoubtedly provide higher quality in the provision of services.

## 6. Conclusions

This document provides an evidence-based overview of research advances and the current state of early intervention. The implementation of systems in early childhood intervention was the basis for the analysis [4].The distribution of journals was analyzed by country, institution, and co-author network, and research topics in the field were identified, including emerging areas for future research. The findings indicate a growing interest in evidence-based practices among international researchers based on a multi-institutional network of co-authors. However, this study has some limitations that should be addressed in future studies. For example, only one database was used in our search, and other document types such as book chapters and conference proceedings were not included. Despite these limitations, the findings of this study provide valuable information for researchers and practitioners in the early intervention field reflecting the state of the published literature rather than research and developments.

## 7. Limitations and Future Research

One of the main limitations of this study is that no specific analysis was found on the construction of systems to support the implementation of evidence-based practices. In the thematic analysis, we obtained clarifying data, since there are research approaches related to “early intervention”, “professional development”, “implementation”, and “coaching or capacity building”, which are topics that are essential to building systems through the implementation of evidence-based practices. The analysis shows us that they are yet to be developed. This is why it would be important to carry out a more in-depth analysis of institutions and co-authorship networks. The work carried out by different researchers can provide relevant information for the development of this field of research.

As we can see in the analysis carried out, no research develops our topic in its entirety, but we do find in all the publications that the subject is related to what we are analyzing. In addition, it is worth mentioning that although the legislative framework (Part C 619 agency) promotes the construction of early care systems and recognized entities such as the ECTA Center and organizations such as Dasy (The Center of IDEA Early Childhood Data Systems), ECPC (Early Childhood Personnel Center), NCPMI (National Center for Pyramid Model Innovations), CDC (Detection and Early Hearing Intervention), NASDSE (National Association of State Directors of Special Education: FY 19 FY 22), and the IDEA Infant & Toddler Coordinators Association, among others, they do not provide the means to implement these systems. After all, this study shows that the construction of systems is scarce when it comes to evidence-based practices [6].

According to our data, we believe that this bibliometric article shows us the importance of creating publications that combine all the elements that a system needs to be able to carry out the adequate implementation of evidence-based practices, producing the best results for children and families. This article can help future researchers to know which are the most discussed topics and to know where the research is going. In addition, we believe it is important to raise awareness of this new field of study to bridge the gap between theory and practice and to ensure that children with disabilities attain the best results possible. It should also be noted that this framework is being applied in the United States and that, thanks to all the practical resources designed, they have a high-quality early care system. Furthermore, it needs to be kept in mind that the early care system must not only consider professional development and the implementation of evidence-based practices, which are topics that are more often researched, but a system consisting of laws, funding, data collection, outcome monitoring, and quality standards is also needed to ensure high-quality services.

## Figures and Tables

**Figure 1 children-10-00813-f001:**
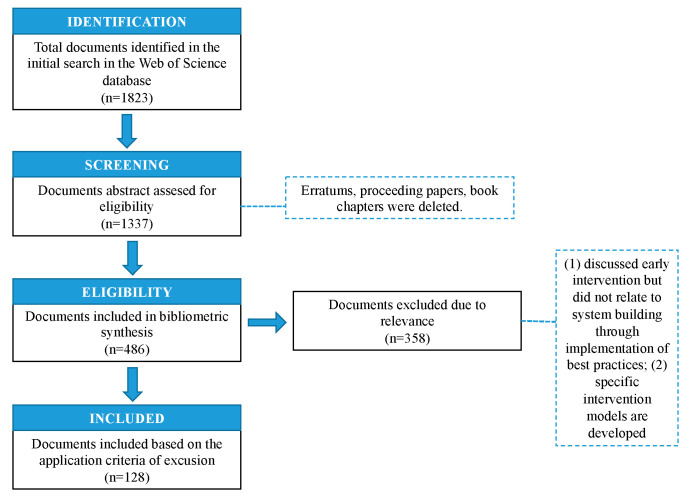
The PRISMA scheme details the steps involved in document identification and selection.

**Figure 2 children-10-00813-f002:**
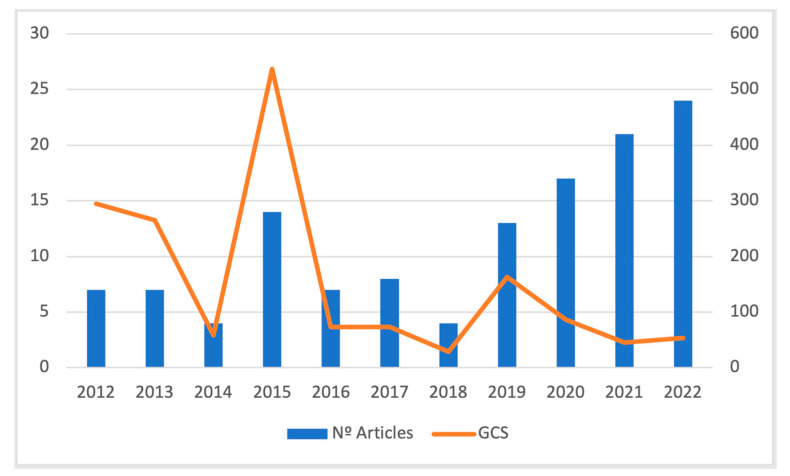
Development of the number of articles and reviews published with the number of global citations over the years (2000–2022).

**Figure 3 children-10-00813-f003:**
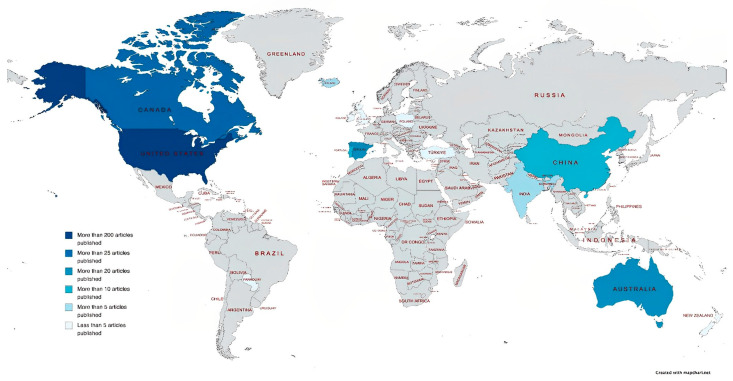
Number of articles published by country.

**Figure 4 children-10-00813-f004:**
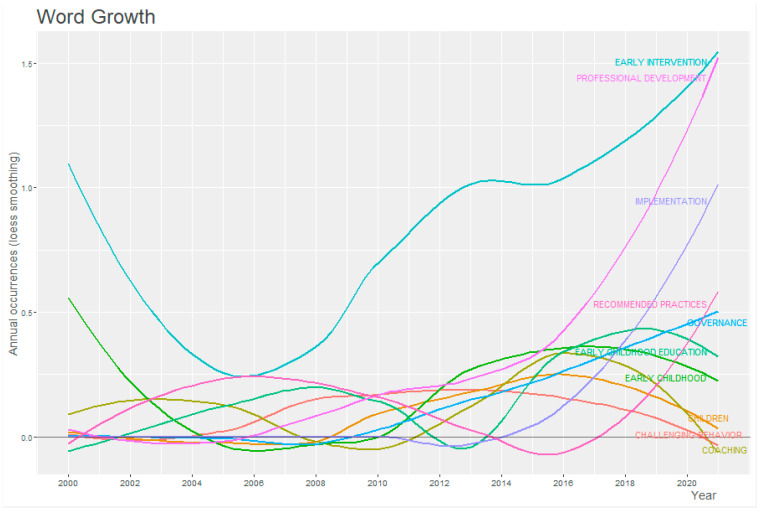
Evolution of keyword usage as a function of year of search.

**Figure 5 children-10-00813-f005:**
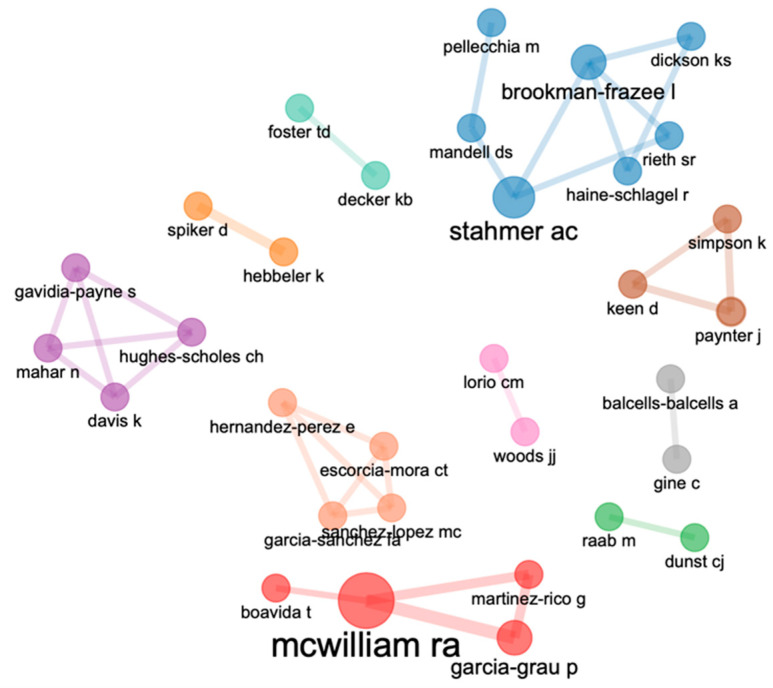
Co-authorship networks.

**Figure 6 children-10-00813-f006:**
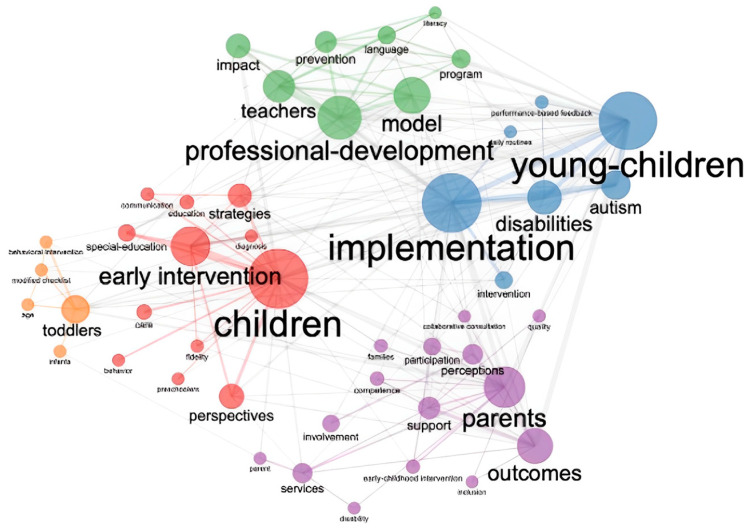
Co-word networks.

**Figure 7 children-10-00813-f007:**
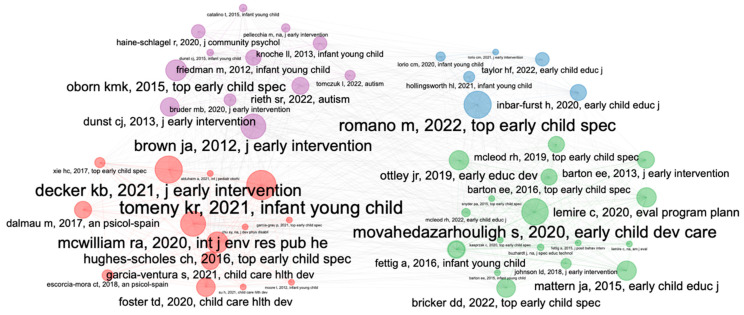
Bibliographic coupling analysis.

**Figure 8 children-10-00813-f008:**
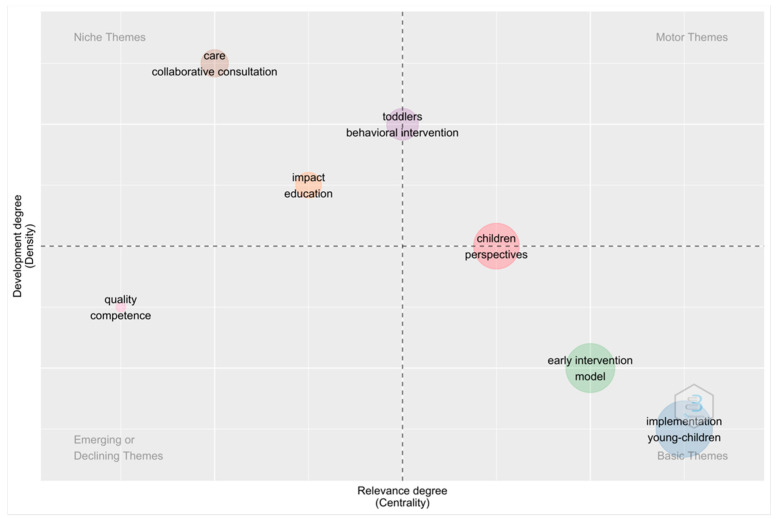
Strategic diagram.

**Table 1 children-10-00813-t001:** Summary information on retrieved evidence-based practices through the creation of effective early care systems.

Main Information about the Data	
Journals	49
Articles	128
Average citations per document	13.11
References	4454
Keywords Plus (ID)	318
Author’s Keywords (DE)	379
Authors	428
Authors of single-authored documents	10
Authors of multi-authored documents	418
Countries	17
Documents per Author	0.29
Authors per Document	3.34
Co-Authors per Documents	4.11
Collaboration Index	3.54

**Table 2 children-10-00813-t002:** Authors with the highest number of publications.

Author	Nb	Institution	LCS	GCS	GCS/Nb
McWilliam, RA	7	University of Alabama	3	41	5.85
Stahmer, AC	6	University of California, Davis	8	184	30.67
Barton, EE	5	Vanderbilt University	6	71	14.2
Snyder, PA	4	University of Florida	15	226	56.5
Dunst, CJ	4	Orelena Hawks Puckett Institute	13	102	25.5
García-Grau, P	4	Catholic University of Valencia	0	27	6.75
Mandell, DS	4	University of Pennsylvania	5	150	37.5
Schnurr, M	4	Iowa State University	0	22	5.5
Brookman-Frazee, L	3	The University of California San Diego	3	43	14.33
Carter, AS	3	University of Massachusetts System	0	22	7.33
428 authors		-	-	-	-

Note: Nb—number of articles; LCS—local citation score; GCS—global citation score.

**Table 3 children-10-00813-t003:** Journals by number of publications and citations received (LCS and GCS) and Impact Factor (JIF).

Journal	Nb	LCS	GCS	GCS/Nb	JIF (2021)
Infants & Young Children	18	40	315	17.5	1.125
Topics in Early Childhood Special Education	15	22	298	19.87	2.313
Journal of Early Intervention	13	14	189	14.54	1.925
Autism	8	7	71	8.87	6.684
Early Childhood Education Journal	7	3	26	3.71	1.656
Child Care Health and Development	6	0	36	6	2.943
Journal of Autism and Developmental Disorders	5	1	188	37	4.345
International Journal of Disability Development and Education	4	0	4	1	1.300
Anales de Psicología	3	4	34	11.33	2.325
American Journal of Evaluation	2	0	26	13	1.507
49 journals		-	-	-	-

Note: Nb—number of articles; LCS—local citation score; GCS—global citation score. JIF—journal Impact Factor.

## Data Availability

Not applicable.

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
