# Peer review of "Bibliometric Analysis on the Implementation of Evidence-Based Practices through Building Effective Systems"

_children, 2023, doi:10.3390/children10050813_

Round 1
Reviewer 1 Report
The introduction does not provide sufficient background about the topic. Improve the contextualization of the theme.
Author Response
Thank you very much for your report. We believe that the corrections made have improved the quality of the manuscript. These considerations have been greatly appreciated in order to improve our background as researchers.
Thank you very much.

Reviewer 2 Report
This manuscript conducted a bibliometric analysis on the implementation of evidence-based practices through building effective systems. A few concerns should be addressed:
(1) Please give some evidence to support WoS “is the most accepted database for the collection and analysis of scientific articles”. One previous study suggested both WoS and Scopus are widely used in academic papers.
(2) I don't quite understand the internal logic of the used search query. Please give more explanation about the search query.
(3) Please explain “Local Citation Score” and “Global Citation Score”.
(4) Table 2: where is “PY”?
(5) Table 3: what does “JCR (2021)” mean?
(6) Line 217: what does “most searched term” mean?
(7) Figure 5: some keywords should be merged (program and programs)
Author Response

(The authors gave the same response as above.)

Reviewer 3 Report
REVIEW COMMENTS
I have only a few concerns about the paper and some suggestions that maybe the authors could consider:
1. To begin with, there are some typos and grammar mistakes. Some long sentences could make readers confused.
2. The type of the paper is not an article; it is reviewing article.
3. The authors should follow the template of the journal, especially in reference style.
4. In the 'Introduction' section, the proposed research gap and the stated objectives do not meet the criteria of proper synergy. Please make the research gap and the research objectives consistent with each other.
5. In Line 25, Page 1, the authors write some non-English words such as “interconnexions”. The author should write each word in English.
6. I think the “Introduction” section can be improved by adding updated references in lines 54-67. I suggest refs. ‘scientometric analysis of scientific literature on neuromarketing tools in advertising ', 'neuromarketing research in the last five years: a bibliometric analysis'. I think these references can help you with the issue.
7. I think the “Materials and Methodology” section can be improved by referencing several articles that have used bibliometric analysis to highlight the importance of bibliometric analysis, which has recently become widely used. I suggest ref. 'current trends in the application of eeg in neuromarketing: a bibliometric analysis' which can be beneficial for this issue.
8. It might be appropriate for the authors to explain why they had chosen the period of extracting data between 2012 and 2022 as described in the “Abstract” and “Materials and Methods” sections.
9. Why have you chosen these specific keywords, i.e., ((early+intervention* OR childhood) AND (implementation OR implementation OR implementation+science OR implementation+practise) OR (system+improvement) OR (recommend+practise)). The authors should explain why they used these keywords in search.
10. The authors should add one more bracket in line 90 at the beginning because the authors use two brackets the last of the search keywords.
11. The author should explain the process of collecting or extracting data clearly for replication. For example, the authors did not refer to what types of documents (e.g., article, book, book chapter, …, etc) they have extracted and what type of documents were eliminated from the WoS database.
12. The authors focused on extracting documents with which languages. The authors should clearly illustrate the language of the extracted documents which were analyzed in this paper.
13. The development of search criteria does not justify why the decisions are made. So, the authors should clarify why they have chosen the WoS database for extraction papers and not the Scopus database or both.
14. Could the authors explain why they have used R-tool and not VOSviewer or both?
15. The authors should list at least the highest 10 productive authors with a minimum number of publications, not only six authors.
16. The authors should list at least the highest 10 productive journals.
17. The authors should explicitly state the novel contribution of this work and its similarities and differences with their previous publications.
18. The authors need to clearly articulate the academic as well as practical implications of this study in a separate section which can be named the theoretical and practical implications of this study.
19. The authors need to clearly articulate the limitations and future research of this study in a separate section which can be named ‘limitations and future research’ behind the conclusion section.
20. For readers to quickly catch your contributions, it would be better to highlight major difficulties and challenges and your original achievements to overcome them in a clearer way in the abstract and introduction.
21. How could/should your study help future studies?
If these revisions can be made to the manuscript, I believe that this study can be accepted for publication.
I wish the authors all the very best with this study.
Author Response

(The authors gave the same response as above.)

Round 2
Reviewer 2 Report
(1) My previous concern about the statement “is the most accepted database for the collection and analysis of scientific articles” has only be partly answered. Please optimize the wording and refer some studies which focus mainly on major databases such as Scopus and Web of Science from authoritative journals in the field of Scientometrics (some case studies focusing on one or two fields are not preferred.)
(2) My previous concern “I don't quite understand the internal logic of the used search query. Please give more explanation about the search query.” has not been responded.
(3) For Table 3, Please replace “JCR (2021)” with “JIF (2021)”
(4) Figure 5: Please confirm that all singular and plural nouns have been merged.
Author Response
Thank you again for your suggestions. With this, the improvement in the document seems substantial, so we have proceeded to address, review, and include each one of them in the final manuscript. We will now proceed to provide a more detailed response to your suggestions:
1) My previous concern about the statement “is the most accepted database for the collection and analysis of scientific articles” has only be partly answered. Please optimize the wording and refer some studies which focus mainly on major databases such as Scopus and Web of Science from authoritative journals in the field of Scientometrics (some case studies focusing on one or two fields are not preferred.)
Thank you for your valuable input. In response to your comment, we would like to reiterate that various studies indicate that Web of Science is the most widely used database for the collection and analysis of scientific articles, as we have previously cited in our first review. However, we acknowledge that both Web of Science and SCOPUS have extensive coverage in the social sciences. Several studies have evaluated and compared different databases, and we provide three examples:
- Bornmann, L. (2014). Do altmetrics point to the broader impact of research? An overview of benefits and disadvantages of altmetrics. Journal of informetrics, 8(4), 895-903.
- Gusenbauer, M., & Haddaway, N. R. (2020). Which academic search systems are suitable for systematic reviews or meta-analyses? Evaluating retrieval qualities of Google Scholar, PubMed, and 26 other resources. Research synthesis methods, 11(2), 181-217.
- Jacso, P. (2005). As we may search-comparison of major features of the Web of Science, Scopus, and Google Scholar citation-based and citation-enhanced databases. Current science, 89(9), 1537-1547.
- Vieira, E.S., Gomes, J.A.N.F. (2009). A comparison of Scopus and Web of Science for a typical university. Scientometrics 81, 587-600 https://doi.org/10.1007/s11192-009-2178-0. https://doi.org/10.1007/s11192-009-2178-0
These articles suggest that the choice of a bibliographic database should depend on the specific needs of the user and the research discipline. They highlight that there is significant overlap between databases, with most articles indexed in several databases. Studies such as Vieira & Gomes (2009) demonstrate that Web of Science provides more accurate bibliographic data in terms of authors, institutional affiliations, and citations. They conclude that around two-thirds of the documents referenced in either of the two databases can be found in both, while the remaining one-third are only referenced in one database.
Considering the purpose of our study, which is to conduct a bibliometric analysis of the subject, we determined that Web of Science is the most appropriate database for our search. However, we acknowledge in the limitations section of our manuscript that it would be interesting to expand the search using databases such as SCOPUS.
We once again appreciate your contribution to the discussion.
(2) My previous concern “I don't quite understand the internal logic of the used search query. Please give more explanation about the search query.” has not been responded.
The search query was developed using keywords extracted from previous research that focused on evaluating the effectiveness of evidence-based practices, as well as methods and techniques for implementing these practices in early care services. The selected keywords were deemed appropriate for understanding the evolution of evidence-based practices in systems building. Some of the studies consulted to identify these keywords include:
- García-Grau, P., Morales-Murillo, C. P., Martínez-Rico, G., Cañadas Pérez, M., & Escorcia Mora, C. T. (2022). Enfoques, prácticas recomendadas, modelos y procedimientos en atención temprana centrados en la familia.Siglo Cero, 53(4), 131-148.
- Loveday, S., Hall, T., Constable, L., Paton, K., Sanci, L., Goldfeld, S., & Hiscock, H. (2022). Screening for adverse childhood experiences in children: a systematic review.Pediatrics, 149(2).
- Vismara, L. A., & Rogers, S. J. (2010). Behavioral treatments in autism spectrum disorder: what do we know?.Annual review of clinical psychology, 6, 447-468.
The aforementioned studies helped identify and select relevant keywords that were included in the search equation. It is important to note that the purpose of the search was to explore the evolution of systems building based on evidence-based practices as a field of study. Therefore, the chosen keywords were relevant to this objective.
(3) For Table 3, Please replace “JCR (2021)” with “JIF (2021)”
Thank you for your observation. This has been modified in the manuscript
(4)Figure 5: Please confirm that all singular and plural nouns have been merged.
Thank you for your observation. This has been revised and modified in the manuscript

Reviewer 3 Report
REVIEW COMMENTS
I have only a few concerns about the paper and some suggestions that maybe the authors could consider:
1. To begin with, there are some typos and grammar mistakes. Some long sentences could make readers confused.
2. The type of paper is not an article, it is reviewing article.
3. I think the “Introduction” section can be improved by adding updated references in lines 74-88. I suggest a refs. 'everything is going electronic, so do services and service quality: bibliometric analysis of e-services and e-service quality', 'a global research trends of neuromarketing: 2015-2020'. I think these references can help you with the issue.
4. The authors should follow the template of the journal, especially in reference style.
5. In the Material and Methods section, the authors have used PRIMA protocol. So, the authors should clearly clarify the process of extracted data for replication. I suggest the refs. 'neuromarketing tools used in the marketing mix: a systematic literature and future research agenda', 'consumer behaviour to be considered in advertising: a systematic analysis and future agenda', which can be useful for this issue. For example, the PRISMA flowchart for selecting publications in this paper.
6. The authors should clearly clarify the type of documents that have been extracted. For example, Table 1 illustrated those articles (What type of articles that authors extracted, i.e., review article, article, ..., etc.).
7. In Table 2, the authors written that the most productive institutions, for example, University of Alabama but the number of published papers is 5.85, could you please explain that how is the number of publications is 5.85?
8. In figure 4, the author should clearly clarify the leading paper of each cluster.
If these revisions can be made to the manuscript, I believe that this study can be accepted for publication.
I wish the authors all the very best with this study.
Author Response
Thank you again for your suggestions. With this, the improvement in the document seems substantial, so we have proceeded to address, review, and include each one of them in the final manuscript. We will now proceed to provide a more detailed response to your suggestions:
1.To begin with, there are some typos and grammar mistakes. Some long sentences could make readers confused.
The document has been sent to the journal's proofreading services for this purpose. The revision of the document will be encouraged. Thank you for your contribution.
- The type of paper is not an article, it is reviewing article.
The journal will be notified to make the appropriate change. Thank you very much.
- I think the “Introduction” section can be improved by adding updated references in lines 74-88. I suggest a refs. 'everything is going electronic, so do services and service quality: bibliometric analysis of e-services and e-service quality', 'a global research trends of neuromarketing: 2015-2020'. I think these references can help you with the issue.
The introduction has been modified to include the references you have pointed out. Thank you for your contribution. This further substantiates the usefulness of this type of analysis.
- The authors should follow the template of the journal, especially in reference style.
The references of this document have been reviewed.
- In the Material and Methods section, the authors have used PRIMA protocol. So, the authors should clearly clarify the process of extracted data for replication. I suggest the refs. 'neuromarketing tools used in the marketing mix: a systematic literature and future research agenda', 'consumer behaviour to be considered in advertising: a systematic analysis and future agenda', which can be useful for this issue. For example, the PRISMA flowchart for selecting publications in this paper.
A diagram showing the PRISMA process for the selection of the publications analyzed in this study is included in the material and method section.
- The authors should clearly clarify the type of documents that have been extracted. For example, Table 1 illustrated those articles (What type of articles that authors extracted, i.e., review article, article, ..., etc.).
Only articles were considered for the document selection process. For this purpose, documents such as proceeding papers, books chapters, etc. were excluded from the search. This is shown in line 126 of the manuscript as well as in the inserted flow chart.
- In Table 2, the authors written that the most productive institutions, for example, University of Alabama but the number of published papers is 5.85, could you please explain that how is the number of publications is 5.85?
The data reported in Table 2 refer to the information extracted from the most relevant authors in the search. Regarding the author McWilliam, formerly of the University of Alabama, he has a total of 7 publications in the search performed. The GCS/Nb index is a bibliometric indicator used to evaluate the scientific productivity of a researcher in a specific field of research. It is calculated by dividing the total number of citations of the researcher's articles in each bibliographic database by the total number of articles he/she has published resulting from the search in that database. Thus, the GCS/Nb index measures the average number of citations that each article published by a researcher receives as a function of the search performed. It is important to bear in mind that this bibliometric indicator only applies to articles indexed in the specific bibliographic database used to calculate it and does not represent the researcher's entire scientific production.
- In figure 4, the author should clearly clarify the leading paper of each cluster.
Figure 4 refers to coauthorship analysis. A coauthorship analysis is a bibliometric method used to study scientific collaborations between authors in each research area. This analysis is performed by identifying the co-authors of a set of scientific publications and analyzing the frequency and pattern of these collaborations. In a coauthorship analysis, different bibliometric measures can be obtained to evaluate scientific collaborations, such as the number of joint publications between authors, the number of coauthors per publication, the number of citations received by collaborative publications, among others, but it does not determine whether one of the articles has greater relevance with respect to other manuscripts in the same group. With this analysis, scientific collaborations between authors are represented in the form of co-authorship networks. In these networks, each author is represented as a node and the collaborations between them are represented by links or connections. These networks can be useful to identify groups of researchers working together in a specific area among other metrics.
